# Assessment of Biodegradation Mechanisms of Ceftiofur Sodium by *Escherichia* sp. CS-1 and Insights from Transcriptomic Analysis

**DOI:** 10.3390/microorganisms13061404

**Published:** 2025-06-16

**Authors:** Meng-Yang Yan, Cai-Hong Zhao, Jie Wu, Adil Mohammad, Yi-Tao Li, Liang-Bo Liu, Yi-Bo Cao, Xing-Mei Deng, Jia Guo, Hui Zhang, Hong-Su He, Zhi-Hua Sun

**Affiliations:** 1State International Joint Research Center for Animal Health Breeding, College of Animal Science and Technology, Shihezi University, Shihezi 832000, China; yanmengyang0707@163.com (M.-Y.Y.); z7142417@163.com (C.-H.Z.); adilswabi15@gmail.com (A.M.); 20222313102@stu.shzu.edu.cn (Y.-T.L.); llb@shzu.edu.cn (L.-B.L.); 20242113056@stu.shzu.edu.cn (Y.-B.C.); 20182313005@stu.shzu.edu.cn (X.-M.D.); guojia@stu.shzu.edu.cn (J.G.); prof.zhang@foxmail.com (H.Z.); 2Development Service Center for Livestock and Aquatic Products of Shihezi City, Shihezi 832000, China; 18997888015@163.com

**Keywords:** ceftiofur sodium, *Escherichia coli* CS-1, degradation pathway, biodegradation mechanisms

## Abstract

Ceftiofur sodium (CFS) is a clinically significant cephalosporin widely used in the livestock and poultry industries. However, CFS that is not absorbed by animals is excreted in feces, entering the environment and contributing to the emergence of antibiotic-resistant bacteria (ARB) and antibiotic-resistant genes (ARGs). This situation poses substantial challenges to both environmental integrity and public health. Currently, research on the biodegradation of CFS is limited. In this study, we isolated a strain of *Escherichia coli*, designated *E. coli* CS-1, a Gram-negative, rod-shaped bacterium capable of utilizing CFS as its sole carbon source, from fecal samples collected from hog farms. We investigated the effects of initial CFS concentration, pH, temperature, and inoculum size on the degradation of CFS by *E. coli* CS-1 through a series of single-factor experiments conducted under aerobic conditions. The results indicated that *E. coli* CS-1 achieved the highest CFS degradation rate under the following optimal conditions: an initial CFS concentration of 50 mg/L, a pH of 7.0, a temperature of 37 °C, and an inoculum size of 6% (volume fraction). Under these conditions, *E. coli* CS-1 was able to completely degrade CFS within 60 h. Additionally, *E. coli* CS-1 exhibited significant capabilities for CFS degradation. In this study, six major degradation products of (CFS) were identified by UPLC–MS/MS: desfuroyl ceftiofur, 5-hydroxymethyl-2-furaldehyde, 7-aminodesacetoxycephalosporanic acid, 5-hydroxy-2-furoic acid, 2-furoic acid, and CEF-aldehyde. Based on these findings, two degradation pathways are proposed. Pathway I: CFS is hydrolyzed to break the sulfur–carbon (S–C) bond, generating two products. These products undergo subsequent hydrolysis and redox reactions for gradual transformation. Pathway II: The β-lactam bond of CFS is enzymatically cleaved, forming CEF-aldehyde as the primary degradation product, which is consistent with the biodegradation mechanism of most β-lactam antibiotics via β-lactam ring cleavage. Transcriptome sequencing revealed that 758 genes essential for degradation were upregulated in response to the hydrolysis and redox processes associated with CFS. Furthermore, the differentially expressed genes (DEGs) of *E. coli* CS-1 were functionally annotated using a combination of genomics and bioinformatics approaches. This study highlights the potential of *E. coli* CS-1 to degrade CFS in the environment and proposes hypotheses regarding the possible biodegradation mechanisms of CFS for future research.

## 1. Introduction

Antibiotics are used to treat and prevent diseases in both humans and animals, and they are extensively employed in farming and agriculture [1]_._ However, antibiotics cannot be effectively absorbed by the body and are difficult to degrade, leading to their accumulation in the environment [2,3]. The persistence of antibiotics in the environment depends on their physical and chemical properties, which affect their degradation rates and environmental behavior. The large-scale use of veterinary drugs has led to the frequent detection of veterinary drug residues in the environment, triggering an increase in the abundance of drug-resistant genes and the accelerated spread of drug-resistant pathogens [4]. It has been suggested that antibiotic resistance genes (ARGs) may pollute the environment more severely than antibiotics themselves [5].

From a practical perspective, it is precisely the abuse of antibiotics in livestock husbandry that has led it to become the primary source of antibiotic accumulation and the spread of antibiotic resistance genes [6]. The co-occurrence of antibiotics, drug-resistant bacteria, and ARGs creates a complex pollution network that poses serious threats to ecosystems and human health [7,8]. Consequently, ARG pollution has become a significant global concern [9].

β-Lactam antibiotics are the highest-selling and most clinically used class of antibiotics on the market, accounting for 70% of the global total antibiotic market share [10]. Cephalosporin antibiotics are a large group of β-lactam antibiotics that are widely used throughout the world. In veterinary medicine, cephalosporins are mainly used for the treatment of bacterial diseases in animals [11]. The extensive use of cephalosporins is the direct cause of their presence in soil, water bodies, and other environments [12]. Compared to human cephalosporins, veterinary cephalosporins have received less research attention regarding their environmental fate and degradation. However, their high usage suggests the potential for environmental contamination, warranting further investigation [13].

CFS is a third-generation cephalosporin antibiotic [14] and is one of the most commonly used antibiotics in veterinary medicine. CFS is widely used to treat bacterial infections in animals; however, a portion of the administered drug, including its bioactive metabolites, is excreted through feces and urine, potentially contaminating the environment and indicating its persistence and possible ecological risks [15].

The widespread use of cephalosporins is the direct cause of their presence in soil, water, and other environments [16].

Microbial treatment is one of the effective ways to degrade cephalosporin residues in the environment. It has received widespread attention because of its advantages of low cost, high degradation efficiency, and minimal secondary contamination. Although some studies have been conducted on the biodegradation of cephalosporins, there is limited information about the specific pathways involved in CFS biodegradation. Moreover, little is known about the key genes and enzymes that mediate this process. Gene sequencing technology, particularly transcriptome sequencing (RNA-Seq), is a powerful tool for studying gene function and structure at a holistic level [17]. It can reveal the molecular mechanisms involved in the microbial degradation of antibiotics.

During the degradation process, CFS-degrading strains express genes associated with antibiotic breakdown when grown in CFS-containing media. Gene annotation of antibiotic-degrading strains provides direct insight into functional genes and enzymes involved in degradation [18], enabling the identification of metabolic pathways and the tagging of key enzymes [19]. These insights help in understanding microbial gene regulation mechanisms in response to antibiotic stress [20]. Despite the increasing interest in antibiotic biodegradation, there have been no studies investigating the biodegradation mechanisms of CFS using transcriptome sequencing technology.

In this study, we isolated a strain of *E. coli*, designated *E. coli* CS-1, from pig manure, which demonstrated a high efficacy in degrading CFS. Based on these findings, we examined the effects of temperature, pH, initial CFS concentration, and inoculum size on the CFS degradation efficiency by *E. coli* CS-1, ultimately identifying the optimal conditions for this process. Furthermore, the degradation products of CFS were characterized using UPLC–MS/MS, and potential degradation pathways were proposed. To delve deeper into the mechanisms underlying CFS biodegradation, we performed transcriptome sequencing of *E. coli* CS-1 to identify the genes involved in this process. This study offers valuable insights into the biodegradation mechanisms of CFS and lays a theoretical foundation for the bioremediation of cephalosporin antibiotic contamination.

## 2. Materials and Methods

### 2.1. Reagents and Media

Ceftiofur sodium (CFS, purity ≥ 98%, Shanghai Yuanye Biotechnology Co., Ltd., Shanghai, China, LOT: S01HS193510) was purchased, while methanol, acetonitrile, and formic acid (≥98%) were obtained as chromatographically pure high-performance liquid chromatography (HPLC) grades from CNW Technologies GmbH (Düsseldorf, Germany). The Luria–Bertani (LB) liquid medium consisted of 10.0 g of peptone, 10.0 g of NaCl, and 5.0 g of yeast extract per liter of distilled water. The LB solid medium was prepared based on the LB liquid medium formula, with the addition of 15–20 g of agar powder. Agar (Guangzhou Saiguo Biotechnology Co., Ltd., Guangzhou, China, Batch number: EZ64AD144) was purchased.

Minimal Salt Medium (MSM, ELITE Bio-tech Co., Ltd., Shanghai, China, LOT: 202402198547) was purchased. It contained 3.5 g/L Na_2_HPO_4_·2H_2_O, 1 g/L KH_2_PO_4_, 0.5 g/L (NH_4_)_2_SO_4_, 0.1 g/L MgCl_2_·6H_2_O, 0.05 g/L Ca(NO₃)_2_·4H_2_O, and 1 mL of micronutrient solution. The micronutrient solution was prepared by dissolving 0.5 g TDTA, 0.2 g FeSO_4_·7H_2_O, 0.01 g ZnSO_4_·7H_2_O, 0.003 g MnCl_2_·4H_2_O, 0.03 g H₃BO₃, 0.02 g CoCl_2_·6H_2_O, 0.001 g CuCl_2_·2H_2_O, 0.002 g NiCl_2_·6H_2_O, and 0.003 g Na_2_MoO_4_·2H_2_O in 1 L of ultrapure deionized water. An additional 20 g/L agar was added to the solid medium. The pH of all media was adjusted to 7.0, and all media were sterilized in an autoclave at 121 °C for 15 min. The half-life of CFS sodium in MSM medium at 37 °C was approximately 15 days, while in wastewater, it exceeded 20 days under the same conditions [21].

### 2.2. Screening and Isolation of CFS-Degrading Bacteria

CFS-degrading bacteria were isolated from fresh manure samples collected from hog farms. The samples were obtained from 500 healthy sows housed in a nursery pig farm. The pigs were approximately 100 days old, with an average body weight of 45–60 kg. All pigs were in good health, and their diet consisted of a commercially balanced ration formulated to meet their nutritional requirements.

A total of 5 g of fecal samples were inoculated into a mineral salt medium (MSM) containing 10 mg/L of CFS. The mixture was placed in a constant-temperature shaker at 37 °C and 180 rpm and incubated for 7 days under light protection. Following the initial 7-day incubation, the culture underwent six enrichment cycles, each lasting 5 days. During these cycles, the CFS concentration was gradually increased from 10 mg/L to 200 mg/L in the following sequence: 10, 25, 50, 100, 150, and 200 mg/L. At the end of each cycle, the cultures were allowed to settle, and the supernatant was discarded. The remaining pellet was then transferred to a fresh MSM medium containing the next concentration of CFS for further adaptation and selection.

After completing the 30-day enrichment process, the cultures were serially diluted with sterile water at ratios of 10^−1^, 10^−2^, 10^−3^, 10^−4^, 10^−5^, and 10^−6^. A volume of 100 μL from each dilution was plated onto a solid MSM medium containing 50 mg/L CFS and incubated at 37 °C for 5 days in the dark.

Single colonies exhibiting different morphologies were selected for plating and isolation. Pure colonies were then inoculated into an MSM medium with a CFS concentration of 100 mg/L. These cultures were shaken at 37 °C and 180 rpm while being protected from light for 72 h to confirm their ability to metabolize CFS. Ultimately, a strain of efficient CFS-degrading bacteria, designated CS-1, was identified and used for subsequent studies. A pure bacterial solution was added to an LB medium containing 40% glycerol and stored frozen at −80 °C for future use [22].

The degradation rate of CFS is calculated as follows:

Degradation rate (%) = 1 − (Ct/C0) × 100%, where C0 is the initial concentration of CFS and Ct is the concentration of CFS at time t [23].

### 2.3. Identification of Strain CS-1

Strain CS-1 was streaked on the plate, and single colonies were observed for color, size, shape, texture, transparency, whether the edges were raised or not, and other morphological characteristics of the colonies. Then, the strains were stained with Gram staining to observe the morphology of the bacteria with an ordinary light microscope. The total DNA of the strain was extracted using a bacterial genomic DNA extraction kit (Majorbio, Shanghai, China), and the bacterial 16S rDNA universal primers (27F: 5′-AGAGTTTGATCCTGGCTCAG-3′, 1492R: 5′-GYTACCTTGTTACGACTT-3′) were used to amplify the 16SrDNA fragments [24]. Beijing Ruiboxingke Biotechnology Co. Ltd., Beijing, China performed the sequencing of 16S rDNA. The sequenced 16S rDNA sequences were submitted to the NCBI database (PQ796435) for sequence comparison, and the 16S rDNA phylogenetic evolutionary tree was constructed using MEGA 11.0 software.

### 2.4. Optimization of Conditions for CFS Degradation by E. coli CS-1

*E. coli* CS-1 was inoculated into the LB liquid medium and cultured with shaking until it reached the logarithmic growth phase. The cells were then collected by centrifugation at 6000 rpm for 10 min, washed three times with sterile PBS solution, and subsequently transferred into the mineral salt medium (MSM) containing CFS for degradation experiments. Basic degradation analysis was conducted in MSM with an initial CFS concentration of 100 mg/L. The culture conditions were as follows: pH 7.0, inoculum size 6%, temperature 35 °C, and agitation at 180 rpm for 72 h in the dark. The effects of varying initial CFS concentrations (25, 50, 100, 150, and 200 mg/L), pH levels (5.0, 6.0, 7.0, 8.0, and 9.0), temperatures (20, 25, 30, 35, and 40 °C), and inoculum sizes (2%, 4%, 6%, 8%, and 10%) (volume fraction).on CFS degradation were investigated using a one-way analysis, with only one parameter varying in each experiment. Each experiment was repeated three times using non-inoculated samples as a blank control.

Following the incubation, the samples were centrifuged at 6000 rpm for 10 min and filtered through a 0.45 μm filter membrane (Merck Millipore Ltd., LOT: RINB54095) to remove macromolecular particles. The collected supernatant was diluted with methanol and incubated at 60 °C for 30 min to denature and precipitate proteins. Finally, the sample was centrifuged again at 8000 rpm for 10 min and filtered through a 0.22 μm membrane (Merck Millipore Ltd., Darmstadt, Germany; LOT: 0000192706). The resulting supernatant was used for HPLC analysis [25].

The residual amount of CFS in the culture broth was determined by high-performance liquid chromatography (HPLC, Agilent Technologies, Lexington, MA, USA). The separation was performed on a ZORBAXSB-A0 column (4.6 mm × 250 mm, 5.0 μm) with a mobile phase of 0.2% trifluoroacetic acid–acetonitrile (70:30, *v*/*v*) at a flow rate of 1.0 mL min^−1^. The column temperature was 40 °C, the injection volume was 20 μL, and a UV detector monitored the process at 290 nm.

Optimal one-factor conditions were combined, and the biodegradation experiments were carried out at 180 rpm and under light protection. The degradation rate of CFS was detected every 3 h at the beginning of the experiment (first 12 h) and every 12 h at the end of the experiment (12 h–72 h), and the incubation process lasted for 3 days.

### 2.5. Identification of Biodegradable Products of E. coli CS-1

The intermediates involved in the degradation of CFS by *E. coli* CS-1 were characterized using ultra-performance liquid chromatography–tandem mass spectrometry (UPLC-MS/MS). *E. coli* CS-1, cultured to its logarithmic growth phase, was centrifuged to collect the bacterial cells. These cells were then added to MSM with an initial CFS concentration of 100 mg/L, adjusted to pH 7.0, and inoculated at a size of 6% (volume fraction). The mixture was incubated at 37 °C under light protection. Three sets of parallel experiments were conducted, along with a set of blank controls that were not inoculated with the *E. coli* CS-1 strain. Samples were collected at three different stages of degradation: early (12 h), middle (24 h), and late (36 h).

For analysis, 1 mL of each sample was pre-treated, filtered through a 0.22 μm organic filter membrane, and collected in a brown injection bottle. A 10 μL aliquot was then separated using an HSS T3 column (100 mm × 2.1 mm i.d., 1.8 μm) prior to mass spectrometric detection. Mobile phase A consisted of a water/acetonitrile (95/5) solution containing 0.1% formic acid, while mobile phase B was a mixture of acetonitrile/isopropanol/water (47.5/47.5/5) also containing 0.1% formic acid. The flow rate was maintained at 0.40 mL/min at a column temperature of 40 °C. Mass spectrometry signals from the samples were acquired in both positive and negative ion scanning modes within a mass scanning range of 70–1050 *m*/*z*. The sheath gas flow rate was set to 50 psi, and the auxiliary gas flow rate was set to 15 psi, with the auxiliary gas heated to 400 °C. The capillary temperature was maintained at 350 °C, with a positive-mode ion spray voltage of 3400 V and a negative-mode ion spray voltage of −2800 V. The normalized collision energy was set to cyclic collision energies of 20, 40, and 60 eV.

The primary mass spectrum resolution was 60,000, while the resolution for the secondary mass spectrum was 15,000, with data collected in Data-Dependent Acquisition (DDA) mode. Upon completion of the analysis, the UPLC–MS raw data were imported into the metabolomics processing software Progenesis QI 3.0 (Waters Corporation, Milford, MA, USA) for baseline filtering, peak identification, integration, retention time correction, and peak alignment. This process resulted in a data matrix comprising retention times, mass-to-charge ratios, and peak intensities. Concurrently, the mass spectral information was matched with public databases to obtain metabolic information on the metabolites.

### 2.6. Transcriptome Sequencing of E. coli CS-1

#### 2.6.1. Sample Handling

The cultured *E. coli* CS-1 bacterial solution was centrifuged at 5000 r/min for 10 min, and the supernatant was discarded. The bacterial pellet was washed with sterile PBS and resuspended, followed by another centrifugation to remove the supernatant. This washing process was repeated three times before collecting the bacterial pellet. Bacteria were then added to the MSM liquid medium containing 50 mg/L CFS and incubated under light protection to achieve a final concentration of 1 × 10^8^ CFU/mL for the *E. coli* CS-1 culture. Immediately after, 1 mL of the bacterial solution was taken, centrifuged, and filtered. The collected organisms were used for transcriptome analysis, while organisms cultured in the medium without CFS under the same incubation conditions served as the control. Three replicates were established for both the experimental and control groups.

#### 2.6.2. Transcriptome Analysis

Total RNA was extracted using the RNAprep Pure Cell/Bacteria kit(Tiangen, Beijing, China) and the RNA concentration of each sample was determined using a Thermo Fisher NanoDrop UV–Vis spectrophotometer (Thermo Fisher, Waltham, MA USA). Ribonucleic acid (rRNA) was removed using the RiboCop rRNA Depletion Kit for Mixed Bacterial Samples (Lexogen, Greenland, NH, USA). The mRNA was randomly fragmented into small pieces of approximately 200 bp, and the fragmented RNA was subsequently converted into complementary DNA (cDNA) to construct a cDNA library. Finally, the genes were sequenced using the Illumina NovaSeq 6000 platform, and transcriptome data were sequenced by Shanghai Meiji Biotechnology Co., Ltd. (Shanghai, China). The ends of the read with low sequencing quality (quality scores less than Q20) were trimmed to obtain high-quality reads. A gene is considered a differentially expressed gene (DEG) if it meets both of the following criteria: a false discovery rate (FDR) of less than 0.05 and an absolute log2 fold change (|log2FC|) greater than 1.

#### 2.6.3. qRT-PCR Validation

Since sequencing is based on bioinformatics methods to calculate the gene expression levels, there is a certain margin of error. To ensure the accuracy of the detection results, the expression trends of the genes need to be further validated by qRT-PCR. The primers for qRT-PCR were synthesized by Meiji Biomedical Technology (Shanghai) Co., Ltd. (Shanghai, China), and the primer information can be found in Table 1, where the internal reference gene is puuD. The reaction system and reaction conditions are shown in Table 2 and Table 3, respectively.

## 3. Results and Discussion

### 3.1. Screening of CFS-Degrading Strains

Nine strains of bacteria capable of degrading CFS were isolated from fresh fecal sludge collected from pig farms. The degradation rates of CFS by these nine strains are shown in Figure 1. Among them, strain 6 exhibited the highest degradation rate, exceeding 90%. In comparison, the other eight strains displayed varying degradation rates, all below 60%. Consequently, strain 6 was selected for subsequent experimental studies and designated as CS-1.

### 3.2. Identification of Bacterium CS-1, Highly Efficient in Degradation

CFS Strain CS-1 exhibited remarkable efficiency in degrading over 90% of 50 mg/L of CFS within three days and was capable of utilizing CFS as its sole carbon source. On the screening medium, CS-1 colonies appeared translucent, slightly convex, and round with regular edges and displayed smooth surfaces with vigorous growth (Figure 2a). Microscopic examination following Gram staining revealed that CS-1 is a Gram-negative, rod-shaped bacterium (Figure 2b).

The 16S rDNA gene sequence of strain CS-1 was submitted to the NCBI database and analyzed using BLAST (http://blast.ncbi.nlm.nih.gov/Blast.cgi, accessed on 11 June 2025) to identify homologous sequences. Phylogenetic analysis using MEGA 11.0 software (Figure 2c) showed that strain CS-1 has 99.217% homology with *E. coli* CP024851.1. Based on colony morphology, Gram microscopy results, and gene sequence alignment, strain CS-1 was identified as *E. coli*.

It is hypothesized that *E. coli* CS-1 may not be a dominant intestinal microorganism in the porcine intestinal tract. Its effective degradation of CFS was likely enhanced through laboratory domestication using selective media designed to favor its growth and promote its CFS degradation capacity. This suggests that the strain was selectively adapted to utilize CFS as a nutrient source, thereby enhancing its metabolic efficiency under controlled conditions.

The identification of *E. coli* CS-1 as a highly efficient CFS degrader highlights its potential for antibiotic bioremediation. Members of the *Enterobacteriaceae* family, including *E. coli*, have previously demonstrated the ability to degrade a variety of antibiotics and other organic pollutants in environmental settings. For instance, *E. coli* HS21, isolated from marine environments, was capable of degrading 66% of sulfapyridine within two days [26]. Similarly, *Klebsiella* sp. CLX-3, isolated from activated sludge, exhibited the ability to degrade 99% of cephalosporin within 12 h at a concentration of 10 mg/L [27].

The present study aligns with findings from previous research demonstrating the bioremediation potential of *Enterobacteriaceae* strains. *Serratia marcescens* WW1, isolated from wastewater treatment plants, showed 89.5% degradation of tetracycline at 20 mg/L within 48 h [28], while *Klebsiella oxytoca* TYL-T1, isolated from soil, efficiently degraded 99.34% of tylosin within 36 h [29]. These examples further reinforce the metabolic adaptability of this bacterial group in degrading organic contaminants.

Apart from antibiotics, *E. coli* has been reported to degrade other environmental pollutants, including polypropylene (PP) [30] 2,4-dinitrotoluene (2,4-DNT) [31], and 4-fluorophenol (4-FP). This highlights *E. coli*’s metabolic flexibility and genetic plasticity, enabling it to target structurally diverse compounds, including CFS, through pathways involving hydrolysis, oxidation, and reduction reactions.

Despite the dominance of *E. coli* CS-1 under laboratory conditions, its performance under natural environmental settings may differ due to the presence of competing microbial species and variations [32] in substrate availability. Nevertheless, the adaptability of *E. coli* to nutrient-depleted environments and its ability to form biofilms could provide a competitive advantage for antibiotic degradation in contaminated ecosystems.

The efficiency of *E. coli* CS-1 is comparable to other antibiotic-degrading strains, such as *Bacillus cereus* P41, which rapidly degraded CFS isolated from the feces of untreated cows [33]. Similarly, Rafii et al. [14] isolated 21 bacterial strains from bovine feces, all capable of degrading CFS at concentrations ranging from 1 to 32 mg/mL, emphasizing the widespread presence of CFS-degrading bacteria in animal waste.

In contrast, fungal strains such as *Ustilago* sp. SMN03 demonstrated 81% degradation of cefdinir at a concentration of 200 mg/L within 6 days [34]. Similarly, *Pseudomonas* spp. CE21 and CE22, isolated from activated sludge, exhibited the ability to degrade cefadroxil, with CE22 achieving 90% degradation within 24 h [35]. These findings highlight the diverse microbial pathways available for antibiotic degradation and underscore the potential for microbial consortia to enhance bioremediation efficiency.

The identification of *E. coli* CS-1 adds to this growing body of evidence, demonstrating that laboratory domestication can effectively enhance CFS degradation. The strain’s adaptability and ability to utilize CFS as a sole carbon source make it a promising candidate for bioremediation strategies aimed at mitigating cephalosporin pollution in agricultural and wastewater systems.

Further studies are required to explore the genetic mechanisms underpinning CFS degradation in *E. coli* CS-1, including the roles of hydrolysis enzymes, redox reactions, and transport proteins. Additionally, evaluating its performance in complex microbial communities and field-scale bioremediation experiments will be critical for validating its application in environmental management practices.

### 3.3. Optimization of Degradation Conditions of E. coli CS-1

To investigate the impact of environmental conditions on the degradation of CFS by *E. coli* CS-1, we manipulated various factors, including temperature, pH, initial CFS concentration, and the amount of bacterial inoculum. We then compared the degradation rates of CFS in MSM medium to identify the optimal conditions for degradation by *E. coli* CS-1. Generally, temperature is a critical factor influencing the microbial degradation of antibiotics, and previous studies have confirmed that an appropriate temperature enhances microbial growth and metabolic activity [36]. In this study, *E. coli* CS-1 exhibited the highest CFS degradation rate at 35 °C, reaching 99.87%. In contrast, at 20 °C, the degradation rate was only 76.99% (Figure 3a). It is hypothesized that higher temperatures within this range are more conducive to the growth of *E. coli* CS-1. At lower temperatures, the activities of some enzymes involved in CFS degradation were reduced, leading to a decreased degradation rate.

Furthermore, pH is one of the critical factors influencing the biodegradation rate of environmental pollutants [37]. In this study, *E. coli* CS-1 exhibited the highest degradation rate of CFS at 35 °C, reaching 99.87%. In contrast, at 20 °C, the degradation rate was only 76.99% (Figure 3a). It is hypothesized that higher temperatures within this range are more conducive to the growth of *E. coli* CS-1. At lower temperatures, the activities of some enzymes involved in CFS degradation were reduced, leading to a decreased degradation rate. Furthermore, pH is one of the critical factors influencing the biodegradation rate of environmental pollutants [38]. The highest CFS degradation rate by *E. coli* CS-1 was observed at pH 7, reaching over 99%. In contrast, the degradation rate was only 63% at pH 5. Additionally, the degradation rate was higher under alkaline conditions compared to acidic conditions (Figure 3b). These findings suggest that unsuitable pH levels can adversely affect enzyme activity, as well as the bacteria’s nutrient uptake and utilization, ultimately inhibiting the growth and metabolism of the strain [39].

Compared to temperature and pH, the effect of CFS concentration on the degradation rate was somewhat reduced. Within the range of CFS concentrations from 25 to 200 mg/L, *E. coli* CS-1 exhibited improved growth, resulting in a degradation rate of CFS exceeding 80%. The degradation rate peaked at 99.49% when the CFS concentration was 50 mg/L. However, it decreased as the CFS concentration increased (Figure 3c). It is speculated that the rise in antibiotic concentration may lead to the accumulation of various intermediates in the medium, which could negatively impact the growth of the strains [20]. Moreover, other studies have also observed a decrease in degradation rate with increasing substrate concentration. Dong et al. [20] investigated the degradation characteristics of *Bacillus cereus* H38 in the degradation of sulfadimethoxine. They found that the degradation efficiency of sulfadimethoxine by H38 decreased as the initial concentration of sulfadimethoxine increased. The amount of inoculum had the least impact on the degradation rate compared to other factors. The optimal inoculum concentration of *E. coli* CS-1 for the degradation of CFS was found to be 6%, resulting in a degradation rate of 99.59 under this condition. However, as the inoculum quantity continued to increase, the degradation rate decreased (Figure 3d). This decline is attributed to excessive inoculum limiting the availability of oxygen and nutrients necessary for microbial growth [40] and the microorganisms competing for the limited nutrients and entering the decline period rapidly [41]. Additionally, *E. coli* CS-1 exhibits strong environmental adaptability, demonstrating a degradation rate of CFS exceeding 60% under various conditions, including low temperatures (20 °C), high temperatures (40 °C), acidic conditions (pH 5.0), alkaline conditions (pH 9.0), and high concentrations of CFS (200 mg/L). This further underscores the potential value of *E. coli* CS-1 for environmental remediation.

In summary, the optimal conditions for CFS degradation by *E. coli* CS-1 were identified as a temperature of 35 °C, a pH of 7.0, an initial CFS concentration of 50 mg/L, and an inoculum concentration of 6%. Under these conditions, the residual CFS amount was continuously reduced, achieving the fastest degradation rate between 12 and 36 h, with complete degradation of CFS occurring within 60 h (Figure 3e).

### 3.4. Analysis of CFS Degradation Products and Degradation Pathways

The analysis of degradation products and pathways is crucial for understanding the biodegradation mechanisms of antibiotics [28]. Typically, it is the specific enzymes produced by microorganisms that facilitate the biodegradation of antibiotics by altering their structures through modification or hydrolysis [42]. In this study, we identified six major degradation products resulting from the degradation of Cefsulodin (CFS) using UPLC-MS/MS (Figure 4): Desfuroyl Ceftiofur (P1), 5-hydroxymethyl-2-furaldehyde (P2), 7-Aminodesacetoxycephalosporanic acid (P3), 5-Hydroxy-2-furoic acid (P4), 2-Furoic Acid (P5), and CEF-aldehyde (P6). These findings suggest two possible pathways for the degradation of CFS. Pathway I: Initially, CFS is hydrolyzed to produce ceftiofur (*m*/*z* = 1047.06), which breaks the sulfur–carbon (SC) bond during the hydrolysis reaction, resulting in the formation of P1 (*m*/*z* = 430.03) and P2 (*m*/*z* = 127.04). Upon further hydrolysis, the carbon–nitrogen (CN) bond in P1 is cleaved, leading to the formation of P3 (*m*/*z* = 215.05). Subsequently, P2 undergoes a redox reaction, producing P4 (*m*/*z* = 173.01), which is then converted to P5 (*m*/*z* = 113.02) through the removal of a hydroxymethyl group via the cleavage of a carbon–carbon (CC) bond. Pathway II: Previous studies have indicated that the biodegradation of most β-lactam antibiotics occurs through the cleavage of the β-lactam ring [43]. The degradation pathway of CFS by *E. coli* CS-1 supports this observation. Breakage of the lactam bond of CFS by enzymatic degradation, resulting in the formation of the product P6 (*m*/*z* = 243.10). Furthermore, CEF-aldehyde (P6) is regarded as the primary biodegradation product of CFS [44].

### 3.5. DEGs Transcriptome Analysis

#### 3.5.1. Transcriptomic Characterization

Transcriptomic sequencing was conducted to analyze differentially expressed genes (DEGs) between the CFS-treated strain (*E. coli* CS-1) and the control strain (CK) to identify genes potentially involved in CFS degradation. Genes were selected based on the criteria of log2|FC| ≥ 2 and *p* < 0.05, ensuring that only statistically significant changes in gene expression were considered.

A total of 1497 DEGs were identified, of which 758 genes were significantly up-regulated and 739 genes were significantly downregulated (Figure 5a). These results indicate that CFS exposure exerts a profound influence on the gene expression profile of *E. coli* CS-1, suggesting that the strain activates specific genes to facilitate its metabolic adaptation and degradation capacity in response to antibiotic stress [45].

Gene Ontology (GO) enrichment analysis was performed to categorize the identified DEGs into three functional groups: biological processes (BP), cellular components (CC), and molecular functions (MF). The top 20 GO terms with the most significant differences were analyzed (Figure 5b). Among these categories, the BP group exhibited the highest enrichment, indicating that the metabolic processes are central to the adaptive response of *E. coli* CS-1 to CFS exposure. Enriched BP terms included organic substance metabolism (GO:0071704), primary metabolic processes (GO:0044238), and small molecule metabolic processes (GO:0044281), reflecting the activation of degradation pathways related to carbon and nitrogen metabolism. Significantly enriched CC terms included membrane components (GO:0016020), cytosol (GO:0005829), and cytoplasm (GO:0005737), which suggest the involvement of membrane transport systems and intracellular enzymes in CFS metabolism. In the MF category, significantly enriched terms such as organic cyclic compound binding (GO:0097159) and ion binding (GO:0043167) point to interactions with CFS-like structures that may enable targeted enzymatic reactions. These results support the hypothesis that CFS degradation in *E. coli* CS-1 is closely linked to organic matter metabolism, nitrogen utilization, and compound binding processes, with membrane transporter genes facilitating substrate uptake and efflux mechanisms [46].

KEGG pathway analysis identified the 20 most significantly enriched metabolic pathways (Figure 5c), which were grouped into Metabolism (M) and Environmental Information Processing (EIP) pathways. DEGs associated with the M pathway included genes involved in sulfur metabolism, nitrogen metabolism, amino acid metabolism, and nucleotide metabolism, which are likely responsible for enzymatic hydrolysis and redox reactions during CFS degradation. Genes enriched in the EIP pathway included those involved in ABC transporters, two-component systems, quorum sensing, and biofilm formation, which are essential for substrate transport, stress adaptation, and antibiotic resistance mechanisms. ABC transporter proteins utilize ATP hydrolysis to expel toxic compounds, contributing to CFS degradation and detoxification, while two-component systems enable bacterial adaptation to changing environments, including exposure to antibiotics. Quorum sensing pathways regulate biofilm formation, which can enhance resistance to antibiotics and enzymatic activity during CFS degradation [47].

The enrichment of metabolic pathways indicates that *E. coli* CS-1 employs complex biochemical processes to adapt to CFS exposure and degrade it. Membrane transporters actively expel CFS metabolites, preventing intracellular accumulation and toxicity, while stress adaptation systems allow metabolic reprogramming to enhance survival and degradation efficiency. The activation of quorum sensing pathways further supports the ability of *E. coli* CS-1 to develop resistance mechanisms, which may enhance its ability to withstand antibiotic stress and sustain degradation activity under varying environmental conditions.

The transcriptome analysis revealed that *E. coli* CS-1 responds to CFS exposure by activating genes involved in organic matter metabolism, oxidoreduction, and membrane transport processes, supporting its role in antibiotic degradation. Upregulated DEGs encode enzymes that catalyze hydrolysis and redox reactions, forming the basis of the CFS degradation pathway. The enriched pathways provide insights into the strain’s ability to metabolize CFS while simultaneously activating stress adaptation mechanisms to resist antibiotic toxicity. These findings highlight a dual response strategy—degradation and resistance—enabling the strain to process CFS and survive in antibiotic-contaminated environments efficiently.

#### 3.5.2. CFS Degradation-Related Genes

Table 4 highlights the genes identified for their potential involvement in the degradation of CFS, specifically those demonstrating a fold change (FC) greater than 2.0. The focus was primarily on genes encoding enzymes associated with hydrolysis and redox reactions. Among the notable findings, hydrolase-encoding genes such as Gamma-glutamyl-gamma-aminobutyrate hydrolase (PuuD) and ureidoacrylate amidohydrolase (RutB) exhibited significant upregulation, with PuuD exhibiting an eleven-fold increase in expression.

In addition, genes coding for oxidoreductases showed a substantial increase in expression. For instance, NADP(+)-dependent aldehyde reductase (YghA) showed an 18.7-fold increase in expression, indicating its potential involvement in redox reactions that modify CFS or its intermediates. Similarly, genes encoding 2,5-didehydrogluconate reductase (DkgA), putative oxidoreductase (YohF), glutathione-dependent hydroquinone reductase (YqiG), tartronate semialdehyde reductase 2 (GlxR), and methylglyoxal reductase (DKdA) exhibited fold changes ranging from 9.9 to 18.7, highlighting their probable roles in oxidative stress responses and detoxification.

The substantial upregulation of these genes suggests their functional importance in the degradation pathway of CFS. For example, YqjG (12.9-fold) and GlxR (12.0-fold) are implicated in reactions involving glutathione and aldehyde detoxification, processes often linked to the breakdown of xenobiotics. RutB (7.7-fold), which catalyzes amide hydrolysis, may facilitate the cleavage of CFS derivatives into smaller fragments for further degradation.

These findings are further supported by metabolic pathway data derived from UPLC-MS/MS analysis, which identified intermediate metabolites consistent with enzymatic hydrolysis and redox reactions. The integration of gene expression profiles with metabolite analysis highlights a coordinated mechanism involving multiple enzymes in the degradation process of CFS.

## 4. Conclusions

This study successfully isolated and identified a highly efficient CFS-degrading bacterium, *E. coli* CS-1, from pig manure, demonstrating its potential for bioremediation of CFS contamination. The optimal degradation conditions—temperature (35 °C), pH (7.0), and inoculum size (6%)—enabled *E. coli* CS-1 to completely degrade CFS at an initial concentration of 50 mg/L within 60 h. Metabolite analysis revealed several degradation products and suggested two potential pathways involving hydrolysis and redox reactions. Transcriptome sequencing further identified key genes associated with CFS degradation, including those encoding hydrolases and oxidoreductases. Validation by qRT-PCR revealed that the relative expression levels of the upregulated genes were consistent with the transcriptome sequencing results. The significant expression of these genes provides insights into the molecular mechanisms underlying the degradation process. These findings suggest the potential of *E. coli* CS-1 to efficiently degrade CFS, offering a potential sustainable approach for mitigating antibiotic pollution in agricultural environments. Future studies could focus on validating the function of the genes identified in this research and exploring the microbial consortia involved in CFS degradation. This may help optimize the process further and assess its potential applicability for effective environmental remediation on a larger scale (Figure 6).

## Figures and Tables

**Figure 1 microorganisms-13-01404-f001:**
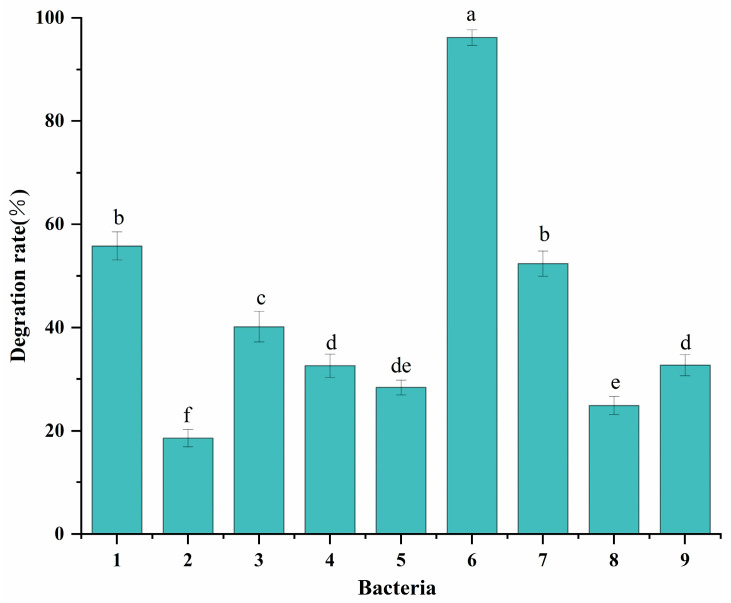
Degradation rates of CFS by different degrading bacteria. Superscript letters (a–e): significant differences among groups (*p* < 0.05).

**Figure 2 microorganisms-13-01404-f002:**
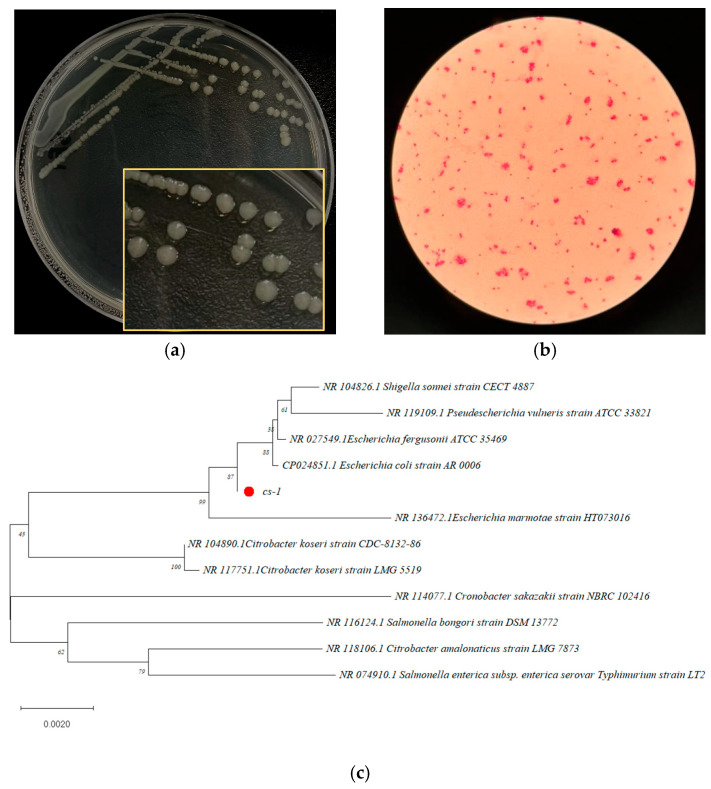
Identification of *E. coli* CS-1. (**a**) Growth morphology of *E. coli* CS-1 on the screening medium; (**b**) Gram micrograph of *E. coli* CS-1 (1000×); (**c**) phylogenetic tree of the CS-1. ● is the sequence of *E. coli* CS-1.

**Figure 3 microorganisms-13-01404-f003:**
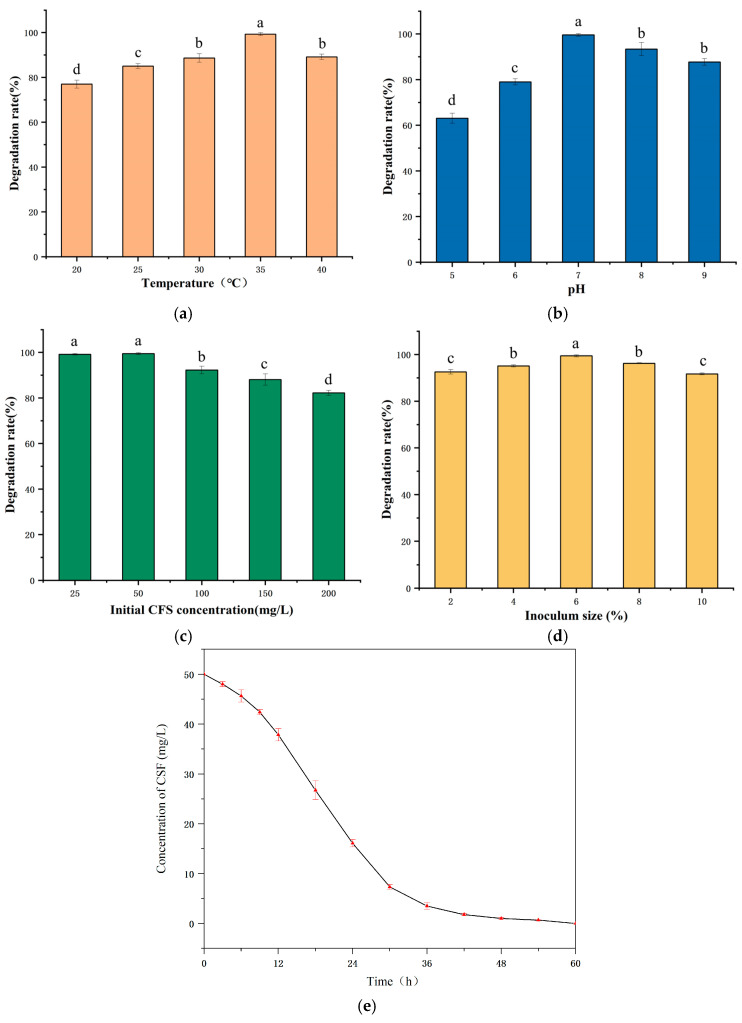
Column chart analysis of the effects of different factors on degradation rate. Effects of (**a**) temperature, (**b**) pH, (**c**) initial CFS concentration, and (**d**) inoculum size on CFS biodegradation by *E. coli* CS-1. (**e**) Degradation of CFS by *E. coli* CS-1 under optimal conditions. Superscript letters (a–d): significant differences among groups (*p* < 0.05).

**Figure 4 microorganisms-13-01404-f004:**
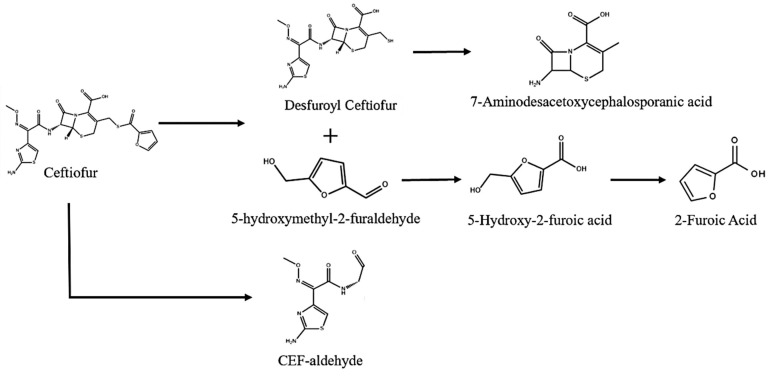
Identification of the main biodegradation products by UPLC-MS/MS and two possible pathways of CFS degradation.

**Figure 5 microorganisms-13-01404-f005:**
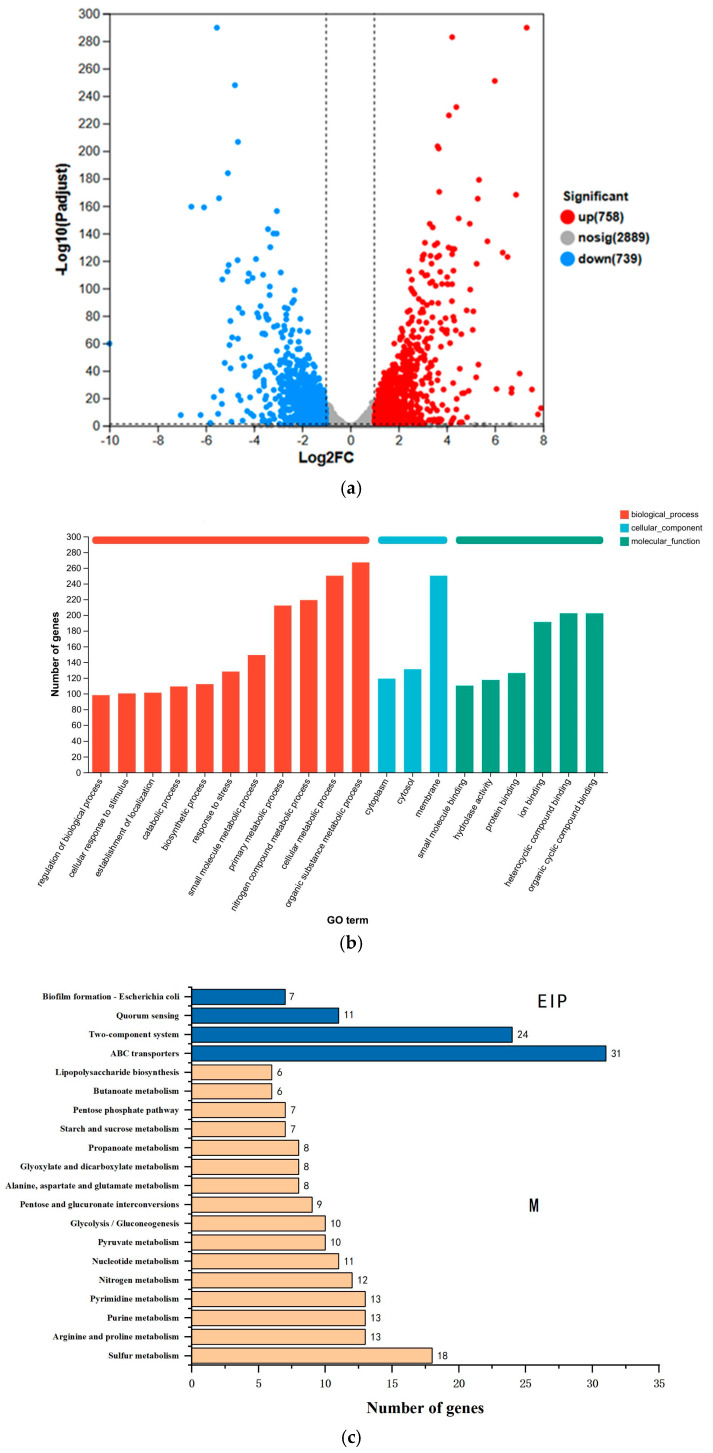
Annotation of *E. coli* CS-1 transcript: (**a**) Differentially expressed gene (DEG) volcano map, (**b**) annotation of the GO function of the *E. coli* CS-1 transcript, (**c**) annotation of the KEGG function of the *E. coli* CS-1 transcript. M: Metabolism pathway; EIP: Environmental Information Processing.

**Figure 6 microorganisms-13-01404-f006:**
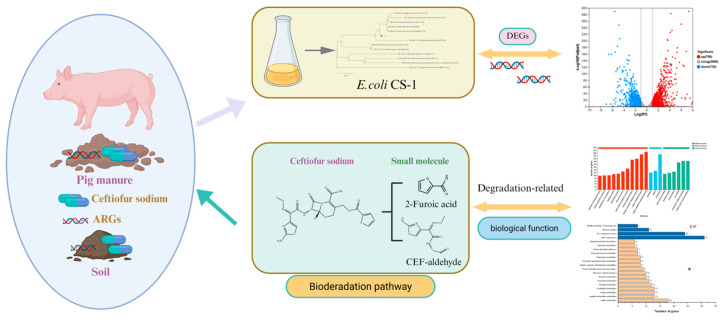
Graphical overview of the biodegradation pathway of ceftiofur sodium by *E. coli* CS-1 associated with pig manure and soil, involving DEGs and degradation-related biological functions.

**Table 1 microorganisms-13-01404-t001:** Quantitative real-time PCR primer sequence information.

Gene Name	Primer Name	Primer Sequence (5′→3′)	Product Length/bp
*lhgD*	*lhgD*-F *lhgD*-R	GTTCCGTGGCGAGTATTT TCCTGAGCGTAGATGGTIT	265
*yghA*	*YghA*-F *YghA*-R	TTTACTGGACTATGCGGCTAC CCAAACTGCGGGATCTTAT	132
*dkgA*	*DkgA*-F *DkgA*-R	TCCGCTGGCATCTGGATAG CGCCGAGTTCGTCTTTGTC	114
*ahr*	*Ahr*-F *Ahr*-R	CCGCAGGCACTGAAAGCAC CGCACCGACCGTATGGAAA	126
*yqjG*	*YqjG*-F *YqjG*-R	CGGTTGGACCTTTGATGAC TGCGGATGATTTCTGCTGATT	182
*PuuD* (Reference gene)	*PuuD*-F *PuuD*-R	CGCATGAAGTTCAGGTTGA AGGATGATTGATGACGCTAA	176

**Table 2 microorganisms-13-01404-t002:** Quantitative real-time PCR reaction system.

Reactant	Concentration	Volume (μL)
2× Taq Plus Master Mix	2X	10
Primer F	5 μM	0.8
Primer R	5 μM	0.8
Template (cDNA)		1
ddH_2_O		7.4
Total		20

**Table 3 microorganisms-13-01404-t003:** Quantitative real-time PCR reaction conditions.

	Temperature	Time	Number of Cycles
Pre-denaturation	95 °C	5 min	1
Denaturation	95 °C	30 s	
Annealing	50 °C	30 s	35
Extension	72 °C	1 min	

**Table 4 microorganisms-13-01404-t004:** Differentially expressed genes (DEGs) in the degrading bacterium *E. coli* CS-1.

Gene Name	Gene Description	FC(c/d)	Log2FC(c/d)	*p*-Value
*lhgD*	L-2-hydroxyglutarate dehydrogenase	5.941	2.570588309	1.21 × 10^−32^
*uxaA*	D-altronate dehydratase	5.807	2.537895134	8.40 × 10^−27^
*astD*	Aldehyde dehydrogenase	4.732	2.242575807	3.49 × 10^−45^
*yghA*	NADP(+)-dependent aldehyde reductase	18.766	4.230028451	1.74 × 10^−81^
*dkgA*	Methylglyoxal reductase (DkgA)	9.968	3.317306509	9.49 × 10^−33^
*yohF*	Putative oxidoreductase YohF	18.735	4.227673213	4.92 × 10^−21^
*puuD*	γ-glutamyl-γ-aminobutyrate hydrolase	11.686	3.546758205	6.80 × 10^−50^
*ahr*	NADPH-dependent aldehyde reductase (Ahr)	5.957	2.574467799	2.93 × 10^−81^
*yqjG*	Glutathionyl-hydroquinone reductase (YqjG)	12.939	3.693655648	3.24 × 10^−77^
*glxR*	Tartronate semialdehyde reductase 2	12.037	3.589455688	2.04 × 10^−24^
*puuD*	γ-glutamyl-γ-aminobutyrate hydrolase	11.686	3.546758205	6.80 × 10^−50^
*rutB*	Ureidoacrylate amidohydrolase	7.695	2.943936191	3.74 × 10^−18^
*norW*	NADH-flavorubredoxin reductase	6.735	2.751693923	4.61 × 10^−16^

FC: Ratio of gene expression between experimental group (c) and control group (d). The *p*-values indicate the significance of the difference.

## Data Availability

The original contributions presented in this study are included in the article. Further inquiries can be directed to the corresponding authors.

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
