# Peer review of "Assessment of Biodegradation Mechanisms of Ceftiofur Sodium by Escherichia sp. CS-1 and Insights from Transcriptomic Analysis"

_microorganisms, 2025, doi:10.3390/microorganisms13061404_

Round 1
Reviewer 1 Report (New Reviewer)
Comments and Suggestions for Authors
The study is a good attempt to provide insights into a novel application of transcriptomics to CFS degradation.
However, there are some corrections to make.
Line 2-3: Authors should modify the title to read “Assessment of Biodegradation Mechanisms of Ceftiofur Sodium by Escherichia sp. CS-1 and insights from Transcriptomic Analysis”
Line 17: Escherichia coli should be in italics.
Line 19: In aerobic or anaerobic condition? Please clarify
Lines 25-26: Please mention the five degradation products and the two degradation pathways proposed based on UPLC-MS/MS analysis.
Line 30: E. coli should be italicized.
Lines 46-47: Kindly adjust this statement “Livestock farming is a major contributor to the accumulation of antibiotics and the transmission of ARGs.” Actually, it’s the abuse of antibiotics in livestock farming that contributes to this.
Line 72: “…70% of the total antibiotic market share” in China or globally? Please clarify the region for this statistic.
Lines 92-95: Add at least one reference for the statements on transcriptome sequencing.
Line 134: for 30 mins? Is this correct, please?
Lines 135-136: Any citations for this information about the half-life?
Figures 1-7: Ensure that figure legends are self-explanatory; currently, some (e.g., Figure 2) lack standalone clarity on experimental conditions.
Very good work, but scientists would like to know the potential field-scale application or validation, which is a key next step.
Authors should confirm whether the Transcriptomic data have been deposited in the NCBI or a similar database. I did not see any statement on this. Why?
Author Response
Comment 1. Line 2-3: Authors should modify the title to read “Assessment of Biodegradation Mechanisms of Ceftiofur Sodium by Escherichia sp. CS-1 and insights from Transcriptomic Analysis”.
Response: Thank you for your suggestion. The authors have agreed to modify the title“Assessment of Biodegradation Mechanisms of Ceftiofur Sodium by Escherichia sp. CS-1 and insights from Transcriptomic Analysis”
Comment 2. Line 17: Escherichia coli should be in italics.
Response: Thank you for pointing this out. The authors have revised it.
Comment 3. Line 19: In aerobic or anaerobic condition? Please clarify.
Response: Thank you for your suggestion. The authors clarify that the condition is anaerobic. Lane 20-22 have revised “We investigated the effects of initial CFS concentration, pH, temperature and inoculum size on the degradation of CFS by strain E. coli CS-1 through a series of single-factor experiments conducted under aerobic conditions. ”
Comment 4. Lines 25-26: Please mention the five degradation products and the two degradation pathways proposed based on UPLC-MS/MS analysis.
Response: Thank you for pointing this out. The authors have add the five degradation products and the two degradation pathways on lane 27-35.
Comment 5. Line 30: E. coli should be italicized.
Response: Thank you for pointing this out. The authors have revised it.
Comment 6. Lines 46-47: Kindly adjust this statement “Livestock farming is a major contributor to the accumulation of antibiotics and the transmission of ARGs.” Actually, it’s the abuse of antibiotics in livestock farming that contributes to this.
Response: Thank you for your valuable suggestion. The authors have accordingly revised the content on lines 55-57.
Comment 7. Line 72: “…70% of the total antibiotic market share” in China or globally? Please clarify the region for this statistic.
Response: Thank you for your suggestion. The authors have clarified that the region in globally, and have revised it on lane 61-62.
Comment 8. Lines 92-95: Add at least one reference for the statements on transcriptome sequencing.
Response: Thank you for your valuable suggestion. The author have add the references.
Comment 9. Line 134: for 30 mins? Is this correct, please?
Response: Thank you for your valuable suggestion. The author have revised it to "for 15 minutes."
Comment 10. Lines 135-136: Any citations for this information about the half-life?
Response: Thank you for pointing this out. The authors have added reference about the half-life.
Comment 11. Figures 1-7: Ensure that figure legends are self-explanatory; currently, some (e.g., Figure 2) lack standalone clarity on experimental conditions.
Response: Thank you for your suggestion. The authors have added the figure legends
Comment 12. Very good work, but scientists would like to know the potential field-scale application or validation, which is a key next step.
Response: Thank you for your suggestion. This is also part of our further plans, we will carry out large-scale applications together after completing related work.
Comment 13.Authors should confirm whether the Transcriptomic data have been deposited in the NCBI or a similar database. I did not see any statement on this. Why?
Response: Thank you for pointing this out. The authors have explained and revised in Data Availability Statement.
Reviewer 2 Report (New Reviewer)
Comments and Suggestions for Authors
The manuscript with the topic "Isolation, Identification, and Degradation Mechanism of Ceftiofur Sodium-Degrading Bacteria Escherichia sp. CS-1 " (Manuscript ID microorganisms-3639377) is written in a professional and scientific language.
I would like to suggest some points to be improved:
- There is an extra comma on the topic which must be deleted after “Identfication”, also on lines 154, 157, 158, 163, 178, 179, 183, 188, 193, 197, 198 twice, 199 twice, 215, 217, 220, 226, 230, 244, 250, 252, 263, 268, 272, 274, 284, 352, 358, 380, 383, 426, 429, 473, 477, 481, 512, 560, 569, 572, 579, 610, 613, 614, 629, 705, 724, 726, 727, 729, 745, 795, 799, 865 (for easier finding “, and”).
- On lines 19, 34, 93, 360, 366, 368 and 720 You have to put Escherichia coli and coli in Italic. Also in Italic Enterobacteriaceae on line 367 and Bacilus cereus on line 390.
- On lines 26, 199, 226 what did You mean with “an inoculum size”? Was it a concentration of bacterial suspension or what? It is not clear.
- All the references in the text must be not superscripted as they were shown on line 44, 49, etc. They must be on normal letter size.
- When You were speaking for the coli designated CS-1 strain I suggest to be written as “E. coli CS-1” on the fowling lines 96, 100, 222, 224, 257, 300, 349, 351, 421, 423, 427, 430, 438, 450, 458, 469, 478, 480, 513, 544, 620, 625.
- Everywhere You mentioned any manufacturer You must put them on the same bracket with the LOT № for example as it is on line 106 and the hall paragraph between 116 to 125 line.
- On line 128 may be was better to be written “waste water” than “fecal water”. If You choose to be “fecal water” explain exactly from which category animals in the farm the samples were- whinning pigs, mothers, mixed from all or else.
- On lines 163 and 262 the dilutions must be superscripted as 10-1.
- On line 184 after the open bracket there was an extra space.
- In my opinion, on lines 185 and 186 the text “Sequencing of 16S rDNA was performed by” have to be deleted and Beijing Ruiboxingke Biotechnology Co. Ltd. (Beijing, China) to be put in one bracket.
- On line 193 is written 6000 rpm for 10 min but on line 203 were 8000 rpm for 10 min. If it was not mistaken, please explain why there was deferens in rpms.
- The text on line 288 must be fixed on the same page with the folowing table.
- Which of both charts is Fig.1?
- The title “3.2. Identification of highly efficient degrading bacteria CS-1” was not understandable; think haw to be changed.
- On line 558 firs must be written “Gene Onthology” and after on brackets “GO”, than to use it in the text.
- Please, explain what is the meaning of abbreviation DEGs?
- 6 to be put before the explanational text.
REFERENCES
27.6% of the references were for a period of the last 5 years. It is mandatory to increase their number.
Author Response
Comment 1. There is an extra comma on the topic which must be deleted after “Identfication”, also on lines 154, 157, 158, 163, 178, 179, 183, 188, 193, 197, 198 twice, 199 twice, 215, 217, 220, 226, 230, 244, 250, 252, 263, 268, 272, 274, 284, 352, 358, 380, 383, 426, 429, 473, 477, 481, 512, 560, 569, 572, 579, 610, 613, 614, 629, 705, 724, 726, 727, 729, 745, 795, 799, 865 (for easier finding “, and”).
Response: Thank you for your valuable suggestion. The authors have revised it.
Comment 2. On lines 19, 34, 93, 360, 366, 368 and 720 You have to put Escherichia coli and coli in Italic. Also in Italic Enterobacteriaceae on line 367 and Bacilus cereus on line 390.
Response: Thank you for pointing this out. The authors have revised it.
Comment 3. On lines 26, 199, 226 what did You mean with “an inoculum size”? Was it a concentration of bacterial suspension or what? It is not clear.
Response: Thank you for highlighting this issue. The authors confirm that the concentration of bacteria refers to the volume fraction of the bacterial suspension, and have accordingly revised the text.
Comment 4. All the references in the text must be not superscripted as they were shown on line 44, 49, etc. They must be on normal letter size.
Response: Thank you for your valuable suggestion. The authors have revised it.
Comment 5. When you were speaking for the coli designated CS-1 strain I suggest to be written as “E. coli CS-1” on the fowling lines 96, 100, 222, 224, 257, 300, 349, 351, 421, 423, 427, 430, 438, 450, 458, 469, 478, 480, 513, 544, 620, 625.
Response: Thank you for your valuable suggestion. The authors have revised it.
Comment 6. Everywhere You mentioned any manufacturer You must put them on the same bracket with the LOT № for example as it is on line 106 and the hall paragraph between 116 to 125 line.
Response: Thank you for your valuable suggestion. The authors have revised it.
Comment 7. On line 128 may be was better to be written “waste water” than “fecal water”. If You choose to be “fecal water” explain exactly from which category animals in the farm the samples were- whinning pigs, mothers, mixed from all or else.
Response: Thank you for your insightful suggestion. The authors have amended the term "fecal water" to "waste water."
Comment 8. On lines 163 and 262 the dilutions must be superscripted as 10-1.
Response: Thank you for your insightful suggestion. The authors have revised it.
Comment 9. On line 184 after the open bracket there was an extra space.
Response: Thank you for your insightful suggestion. The authors have deleted the extra space.
Comment 10. In my opinion, on lines 185 and 186 the text “Sequencing of 16S rDNA was performed by” have to be deleted and Beijing Ruiboxingke Biotechnology Co. Ltd. (Beijing, China) to be put in one bracket.
Response: Thank you for your valuable suggestion. The authors have revised it: (Beijing Ruiboxingke Biotechnology Co. Ltd., Beijing, China) performed the sequencing of 16S rDNA.
Comment 11. On line 193 is written 6000 rpm for 10 min but on line 203 were 8000 rpm for 10 min. If it was not mistaken, please explain why there was deferens in rpms.
Response: Thank you for your valuable suggestion. This was a mistake, and authors have corrected to 6000 rpm on line 216.
Comment 12. The text on line 288 must be fixed on the same page with the folowing table.
Response: Thank you for your insightful suggestion. The author have revised it.
Comment 13. Which of both charts is Fig.1?
Response: Thank you for your valuable suggestion. The authors have revised it.
Comment 14. The title “3.2. Identification of highly efficient degrading bacteria CS-1” was not understandable; think haw to be changed.
Response: Thank you for your insightful suggestion. The authors have revised “3.2. Identification of highly efficient degrading bacteria CS-1” to “3.2 Identification of Bacterium CS-1 Highly Efficient in Degrading CFS”.
Comment 15. On line 558 firs must be written “Gene Onthology” and after on brackets “GO”, than to use it in the text.
Response: Thank you for your insightful suggestion. The authors have revised it.
Comment 16. Please, explain what is the meaning of abbreviation DEGs?.
Response: Thank you for your insightful suggestion.The DEGs refer to differentially expressed genes, a term the authors have elucidated on line 489 since its initial mention.
Comment 17. 27.6% of the references were for a period of the last 5 years. It is mandatory to increase their number.
Response: Thank you for your insightful suggestion.There is limited research on the microbial degradation mechanism of ceftiofur sodium. This study is the first to conduct transcriptomic sequencing on the microbial degradation mechanism of ceftiofur sodium, so there are relatively few relevant references in the past five years.
Reviewer 3 Report (New Reviewer)
Comments and Suggestions for Authors
Dear authors,
Since this is the second revised version of the manuscript, and I can see the substantial changes you have made, I would recommend some minor changes:
Line 19: Escherichia coli should be written in italic
Line 22-23 abbreviation should be used for ceftiofur sodium
Line 34: E. coli should be written in italic
Line 37: I believe that the exact name of the strain that is used would be more appropriate as the keyword
Line 48: Antibiotic resistance is not typically treated as ecotoxicological effect
Lines 59-60: all antibiotics are mainly used to treat bacterial diseases, not only cephalosporins. This sentence shoulb be phrased better
Lines 60-61: Despite their widespread use, 60
cephalosporin residues have been detected in soil, water, and sediments, highlighting their persistence and potential environmental risks. This sentence should be phrased better. Widespread use of cephalosporins is an exact reason why they are present in soil, water etc.
Lines 67-69: The sentence "It inhibits the synthesis of the bacterial cell wall, leading to cell rupture and bacterial death, making it effective in treating infections", should be removed, since the mechanism by which cephalosporins kill bacteria should be well known to the scientific public.
Line 93: Escherichia coli should be written in italic
Line 109: if the city of origin is stated above, this should be followed throughout the manuscript
Lines 106-128: I believe that the whole paragraph about the reagents should be removed. Since the ingredients in microbiological media are unnecessary to be listed, microbiological media and the producers should be simply listed in the following sections (without the information o their composition)
Lines 367-358: E. coli should be written in italic. Please correct this throughout the manuscript
Figure 4: I believe that this figure is unnecessary, since this is not an original result from this study, but the theoretical background on the subject.
Author Response
Comment 1. Line 19: Escherichia coli should be written in italic.
Response: Thank you for your insightful suggestion. All instances of E. coli in the article have been changed to italicized font.
Comment 2. Line 22-23: Use abbreviation for ceftiofur sodium (e.g., CFS)
Response: Thank you for pointing this out. Except for the first occurrence of ceftiofur sodium in the text, all subsequent mentions have been abbreviated.
Comment 3. Line 34: *E. coli* should be written in italic
Response: Thank you for pointing this out. The authors have revised it.
Comment 4. Line 37:I believe the exact name of the strain (e.g., *Escherichia* sp. CS-1) would be more
appropriate as a keyword
Response: Thank you for pointing this out. Based on your feedback and since the strain's genus has been clearly identified, the keywords in the article have been changed to Escherichia coli CS-1.
Comment 5. Line 48: Antibiotic resistance is not typically treated as ecotoxicological effect
Response: Thank you for pointing this out. The authors have deleted the ecotoxicological effect.
Comment 6. Lines 59-60: All antibiotics are primarily used to treat bacterial diseases (not just cephalosporins). This sentence should be rephrased for accuracy.
Response: Thank you for pointing this out. The main purpose of this statement is to introduce the uses of cephalosporins, not to imply that only cephalosporin antibiotics are used to treat bacterial diseases.
Comment 7. Lines 60-61: Despite their widespread use, 60 cephalosporin residues have been detected in soil, water, and sediments, highlighting their persistence and potential environmental risks. This sentence should be phrased better. Widespread use of cephalosporins is an exact reason why they are present in soil, water etc.
Response: Thank you for pointing this out. The authors have revised it on lane 64-65: The extensive use of cephalosporins is precisely the direct cause of their presence in soil, water bodies and other environments.
Comment 8. Lines 67-69: The sentence "It inhibits the synthesis of the bacterial cell wall, leading to cell rupture and bacterial death, making it effective in treating infections" should be removed, as the mechanism of cephalosporins is well-known to the scientific community.
Response: Thank you for pointing this out. The authors have deleted it.
Comment 9. Line 93: *Escherichia coli* should be written in italic .
Response: Thank you for pointing this out. All instances of E. coli in the article have been revised to italicized font.
Comment 10. Line 109: If the city of origin is stated above, this should be consistent throughout the manuscript
Response: Thank you for pointing this out. The authors have revised it.
Comment 11. Lines 106-128: I believe that the whole paragraph about the reagents should be removed. Since the ingredients in microbiological media are unnecessary to be listed, microbiological media and the producers should be simply listed in the following sections (without the information o their composition)Line 109: If the city of origin is stated above, this should be consistent throughout the manuscript
Response: Thank you for pointing this out. The previous review experts' comments indicated that detailed components of the culture medium should be provided to prove that cefotaxime sodium is the sole carbon source.
Comment 12. Lines 367-358: E. coli should be written in italic. Please correct this throughout the manuscript
Response: Thank you for pointing this out. All instances of E. coli in the article have been corrected to italicized font.
Comment 13. Figure 4: I believe that this figure is unnecessary, since this is not an original result from this study, but the theoretical background on the subject.
Response: Thank you for pointing this out. Figure 4 presents the results of this experiment (the two degradation pathways of ceftiofur sodium) in the form of a flowchart, making it more intuitive.
Reviewer 4 Report (New Reviewer)
Comments and Suggestions for Authors
This manuscript addresses an important environmental issue: the biodegradation of veterinary antibiotics, particularly ceftiofur sodium (CFS), by microbial strains. The authors isolate and characterize an Escherichia strain (CS-1) capable of degrading CFS and investigate the degradation mechanism using UPLC-MS/MS and transcriptomic analyses. The work is timely and relevant, especially given increasing concerns about antimicrobial resistance and environmental antibiotic contamination.
It is evident that the authors have made a concerted effort to enhance the quality of their work following a previous round of peer review. However, the manuscript suffers from issues that compromise the quality of the presentation.
Figure legends are not sufficiently informative. Figure legends often restate text rather than providing stand-alone informative captions.
Some conclusions are overstated; the language should be tempered to reflect the evidence. (e.g., Line 713: "These findings underscore the capability... presenting a sustainable approach..." should be softened to reflect potential rather than proven applicability).
It is imperative that italicization of species names (e.g., Escherichia coli) be consistent. Furthermore, the title makes reference to Escherichia spp., whereas in the abstract it is E. coli.
Author Response
Comment 1. Figure legends are not sufficiently informative. Figure legends often restate text rather than providing stand-alone informative captions.
Response: Thank you for pointing this out. The authors have revised it.
Comment 2. Some conclusions are overstated; the language should be tempered to reflect the evidence. (e.g., Line 713: "These findings underscore the capability... presenting a sustainable approach..." should be softened to reflect potential rather than proven applicability).
Response: Thank you for your insightful suggestion. The authors have revised it.
Comment 3. It is imperative that italicization of species names (e.g., Escherichia coli) be consistent. Furthermore, the title makes reference to Escherichia spp., whereas in the abstract it is E. coli.
Response: Thank you for your valuable suggestion. The authors have revised it.
This manuscript is a resubmission of an earlier submission. The following is a list of the peer review reports and author responses from that submission.
Round 1
Reviewer 1 Report
Comments and Suggestions for Authors
This study identifies Escherichia coli strain CS-1, capable of degrading ceftiofur sodium (CFS) from livestock waste. Under optimal conditions, CS-1 completely degrades CFS within 60 hours. UPLC-MS/MS analysis reveals degradation pathways, and transcriptome sequencing highlights key genes involved. The findings suggest CS-1's potential for environmental bioremediation of CFS contamination.
Comments:
1. Please explain the statistical analysis in the method section. Also, explain the mean of "a", "b", "c", "d", "de", "e" in the figure 1 and 3. Did they represent p values? If not, please add the p values for each bar in these figures.
2. Please use the higher resolution version images for figure 5b and 5c.
Reviewer 2 Report
Comments and Suggestions for Authors
This manuscript that entitled by “Isolation, Identification and Degradation mechanism of Ceftiofur sodium-Degrading Bacteria Escherichia sp. CS-1”, investigated the effects of the initial concentration of CFS, pH, temperature, and inoculum size on the degradation of ceftiofur sodium (CFS) by strain CS-1 through one-way experiments. The findings stated that the highest degradation rate of CFS was achieved by E. Coli CS-1 strain under certain environmental conditions. Also, five degradation products were identified, and two degradation pathways were proposed based on UPLC-MS/MS analysis. The manuscript is generally well-addressed and well-cited; however, I have some comments/suggestions.
Line 3: I suggest changing the title to be more concise, specific and relevant. Such as: The biodegradable mechanisms of ceftiofur sodium by isolated Escherichia sp. CS-1strain from pig.
Line 7: Please add number over each above mentioned author and add it also to this affiliation as well.
Line 30: Here is a big concern about how the environmental and governmental agencies will use the E.coli CS-1 to degrade CFS in the environment? how to be applicable? how much does it cost? how about zoonotic importance of using these strain of E.coli? biosecurity standards?
Line 31: I suggest adding CS-1 strain to your keywords as the main one you used it in your study.
Line 33: Please check the spelling of the introduction title.
Line 50: Please note that, by the end of line 50, there is no mention of using the antibiotics at pig farms in China and the previous published trials discussing the using the degradations of antibiotics residues in veterinary field.
Line 65: Until this line no direct mention of previous literature discussing the degradation of any antibiotic residues at any animals or veterinary facility or environmental treatment for them. I think this is the core idea of your conducted study. Please rewrite the introduction to be more specific and focused.
Line 123: Please mention the local or the geographical area where these hog farms are located. Also, add more data about selected animals for collecting the samples.
Line 148: There is no mention to the catalog number for most of kits used in the study. Please revise throughout the manuscript to add it.
Line 247: before adding figure 1, I suggest adding table mentioning the all nine isolated strains with some details to understand the identification of each strain separately.
Line 451: The conclusion is long, please revise to be more focused. It should summarize the key findings of your study, reiterate the main points of your argument, highlight the significance of your research, and discuss potential implications and future research directions while avoiding introducing new information.
Line 452: I think there is no need to mention E Coli between asterisk (*) which is repeated 4 times at the conclusion paragraph.
Line 469: please add figure legend with a little description for it.
Line 483: Please add the approval year for each project mentioned at funding paragraph.
References # 2, 3, 4, 5 and most of them are incomplete. Please revise by following the journal guidelines that’s included: Abbreviated Journal Name Year, Volume, page range. The year should be bold and mentioned before the volume and page numbers.
Comments on the Quality of English LanguageMajor editing of English language required.